# Eukaryotic Translation Initiation Factor 4AI: A Potential Novel Target in Neuroblastoma

**DOI:** 10.3390/cells10020301

**Published:** 2021-02-02

**Authors:** Christina Skofler, Florian Kleinegger, Stefanie Krassnig, Anna Maria Birkl-Toeglhofer, Georg Singer, Holger Till, Martin Benesch, Regina Cencic, John A. Porco, Jerry Pelletier, Christoph Castellani, Andrea Raicht, Ewa Izycka-Swieszewska, Piotr Czapiewski, Johannes Haybaeck

**Affiliations:** 1Diagnostic and Research Center for Molecular BioMedicine, Diagnostic and Research Institute of Pathology, Medical University of Graz, 8010 Graz, Austria; christina.wodlej@medunigraz.at (C.S.); fkleinegger@gmail.com (F.K.); stefanie.krassnig@gmx.net (S.K.); anna.birkl-toeglhofer@i-med.ac.at (A.M.B.-T.); 2Center for Biomarker Research in Medicine, 8010 Graz, Austria; 3Department of Pathology, Neuropathology and Molecular Pathology, Medical University of Innsbruck, 6020 Innsbruck, Austria; 4Department of Pediatric and Adolescent Surgery, Medical University of Graz, 8036 Graz, Austria; georg.singer@medunigraz.at (G.S.); holger.till@medunigraz.at (H.T.); christoph.castellani@medunigraz.at (C.C.); 5Department of Pediatrics and Adolescent Medicine, Division of Pediatric Hematology/Oncology, Medical University of Graz, 8036 Graz, Austria; martin.benesch@medunigraz.at (M.B.); andrea.raicht@klinikum-graz.at (A.R.); 6McIntyre Medical Sciences Building, Department of Biochemistry, McGill University, Montreal, QC H3G 1Y6, Canada; regina.cencic@mcgill.ca (R.C.); jerry.pelletier@mcgill.ca (J.P.); 7Department of Chemistry and Center for Molecular Discovery (BU-CMD), Boston University, Boston, MA 02215, USA; porco@bu.edu; 8Department of Pathology and Neuropathology, Medical University of Gdańsk, 80-211 Gdańsk, Poland; eczis@gumed.edu.pl; 9Department of Pathomorphology, Medical University of Gdańsk, 80-214 Gdańsk, Poland; p.czapiewski@klinikum-dessau.de; 10Department of Pathology, Medical Faculty, Otto von Guericke University Magdeburg, 39120 Magdeburg, Germany; 11Department of Pathology, Klinikum Dessau, 06847 Dessau-Rosslau, Germany

**Keywords:** neuroblastoma, eukaryotic initiation factor 4AI (eIF4AI), rocaglates, CR-1-31-B, SH-SY5Y, Kelly

## Abstract

Neuroblastoma (NB) is the most common extracranial pediatric solid tumor. Children suffering from high-risk and/or metastatic NB often show no response to therapy, and new therapeutic approaches are urgently needed. Malignant tumor development has been shown to be driven by the dysregulation of eukaryotic initiation factors (eIFs) at the translation initiation. Especially the activity of the heterotrimeric eIF4F complex is often altered in malignant cells, since it is the direct connection to key oncogenic signaling pathways such as the PI3K/AKT/mTOR-pathway. A large body of literature exists that demonstrates targeting the translational machinery as a promising anti-neoplastic approach. The objective of this study was to determine whether eIF4F complex members are aberrantly expressed in NB and whether targeting parts of the complex may be a therapeutic strategy against NB. We show that eIF4AI is overexpressed in NB patient tissue using immunohistochemistry, immunoblotting, and RT-qPCR. NB cell lines exhibit decreased viability, increased apoptosis rates as well as changes in cell cycle distribution when treated with the synthetic rocaglate CR-1-31-B, which clamps eIF4A and eIF4F onto mRNA, resulting in a translational block. Additionally, this study reveals that CR-1-31-B is effective against NB cell lines at low nanomolar doses (≤20 nM), which have been shown to not affect non-malignant cells in previous studies. Thus, our study provides information of the expression status on eIF4AI in NB and offers initial promising insight into targeting translation initiation as an anti-tumorigenic approach for NB.

## 1. Introduction

Neuroblastoma (NB) is a pediatric tumor derived from primordial neural crest cells. With an incidence of 10.7 per one million children younger than 15 years, it is the most common extracranial solid tumor in childhood [1]. NB remains distinct from other solid pediatric tumors due to its biological heterogeneity and wide range of clinical behavior. While some children present with tumors regressing completely or differentiating without treatment, many patients suffer highly aggressive metastatic neoplasms that are unresponsive to standard and investigational anti-cancer treatment and are associated with overall survival (OS) rates lower than 50% [2].

Intensive research has revealed dysregulated protein synthesis in cancer cells as a targetable vulnerability of various tumor entities. Cancer cells depend on high protein levels, especially those promoting survival, proliferation, and angiogenesis. In this regard, translation has been shown to be a crucial step in carcinogenesis. Eukaryotic translation is strongly regulated at the initiation step by eukaryotic initiation factors (eIFs). Translation initiation can be modified in cancer cells through various molecular alterations—for example, indirectly via the hyper-activation of oncogenic pathways such as the MAPK/ERK- or PI3K/AKT/mTOR-pathway, as was shown also in NB, or directly via the gain and loss of eIFs and their regulators [3,4,5,6]. The eIF4F complex has been identified as the most important link between carcinogenesis and translation initiation. It is responsible for mRNA guidance to the small ribosomal subunit. The heterotrimeric eIF4F complex consists of eIF4E, which binds the 5′cap structure of mRNAs; eIF4A, an ATP-dependent DEAD-box RNA helicase; and the scaffolding protein eIF4G [7]. eIF4A has two isoforms relevant for the eIF4F complex, eIF4AI and eIF4AII, whereby eIF4AI is the predominant paralog expressed in most cell types. The suppression of eIF4AI cannot be compensated by eIF4AII [8]. Lazaris-Karatzas et al. demonstrated that the overexpression of eIF4E malignantly transforms NIH/3T3 cells in vitro [9], a finding which was later confirmed in mice [10] and was also shown in various cancers, such as breast cancer [11] and hepatocellular carcinoma [12]. Increased expression was also documented for eIF4G and eIF4A—for example, in lung [13] and cervical cancer [14].

Extensive research has been conducted in the field of the screening and development of small-molecule translation inhibitors. A large body of literature exists on natural and synthetic compounds that block protein synthesis at the level of translation initiation. In order not to lose track of these, Dmitriev and colleagues recently published a database called EuPSIC (eukaryotic protein synthesis inhibiting compounds). EuPSIC is a regularly updated database of small-molecule inhibitors that target eukaryotic protein synthesis, currently comprising information on over 350 different compounds (http://eupsic.belozersky.msu.ru) [15]. The question of whether targeting a global process such as protein synthesis is safe frequently arises. By now, one compound directly targeting the protein translation has been approved by the Food and Drug Administration as an antineoplastic approach: omacetaxine mepesuccinate. The semisynthetic compound homoharringtonine, which inhibits the initial elongation step, is approved for patients with chronic myeloid leukemia if they are nonresponding to or failure of tyrosine kinase inhibitors, with some patients exhibiting severe but manageable adverse effects [16,17,18].

Since targeting the translational machinery is clinically relevant and eIF4F complex members are overexpressed in a variety of tumor entities, this is an attractive target for pharmacological anti-neoplastic strategies. Major approaches include the disruption of eIF4E/eIF4G interaction, the disabling of eIF4E interaction with the cap structure, and the inhibition of eIF4A activity (reviewed by Chu and Pelletier [18]). In this regard, eIF4A was reported to be a direct target of rocaglates, a family of small-molecule inhibitors with a cyclopenta[*b*]benzofuran backbone initially isolated from *Aglaia* genus such as silvestrol [19,20]. In the early 2000s, silvestrol was shown to be cytotoxic against a variety of cancer cell lines [21]. The chemical synthesis of silvestrol is challenging. Pelletier, Porco, and colleagues aimed for the chemical synthesis of cyclopenta[*b*]benzofuran (rocaglate) derivatives with reduced complexity. CR-1-31-B (named (-)-**9** in the original work) was demonstrated to be as potent as silvestrol in terms of its inhibition of protein synthesis and cytotoxicity in human Burkitts’ lymphoma BJAB cells [22].

Little is known about the role of dysregulated eIF4F complex members in NB. One study showed a significant positive correlation between immunohistochemical eIF4E staining and age at diagnosis, while other parameters such as OS lacked statistically significant correlation in eIF4E levels [23].

In this study, our main objective was to evaluate the expression of eIF4AI and eIF4E in human NB tissue. Furthermore, we aimed to elucidate the effect of targeting eIF4AI in NB cell lines in order to gain insight into the relevance of using the synthetic rocaglate CR-1-31-B as a potential anti-neoplastic drug in NB.

## 2. Materials and Methods

### 2.1. Human Specimens

The study was reviewed and approved by the ethical committee of the Medical University of Graz (MUG), Graz, Austria (26-420 ex 13/14).

NB cryo-specimens (n = 17) were obtained from the Biobank Graz (MUG), Graz, Austria for biochemical analyses. NB formalin fixed and paraffin embedded (FFPE) material for immunohistochemical (IHC) analyses (n = 36) was provided by E.I-S. Non-neoplastic temporal cortical brain (n = 9) and thoracic spinal cord tissue (n = 9) (NNT) were collected post-mortem at the Diagnostic and Research Institute of Pathology (MUG), Graz, Austria. NNT was used as a control tissue for biochemical and IHC analyses (Table 1).

### 2.2. Cell Lines and Rocaglate CR-1-31-B

The human NB cell line SH-SY5Y was provided by the Core Facility for Alternative Biomodels and Preclinical Imaging (MUG), Graz, Austria. The second human NB cell line, Kelly, was kindly provided by A.R., Division of Pediatric Hematology/Oncology (MUG), Graz, Austria.

The eIF4A inhibitor, CR-1-31-B, was synthesized as described elsewhere [22]. The concentration at which 50% of cells were dead (IC_50_) after 48 h was evaluated using an online tool by AAT Bioquest, Sunnyvale, CA, USA (https://www.aatbio.com/tools/ec50-calculator).

### 2.3. Statistical Analysis

GraphPad Prism version 5.01 (GraphPad Software, San Diego, CA, USA) was used for statistical analysis. The significance level was set at *p* < 0.05. For RT-qPCR and immunoblot data, the Shapiro–Wilk test was used to test for normal distribution, followed by two-tailed unpaired t-test. For immunohistochemical data, the Mann–Whitney *U* test was used. All other data sets were statistically analyzed using a two-way analysis of variance with the Bonferroni post-hoc test.

### 2.4. Immunohistochemistry

Immunohistochemical staining for eIF4AI and eIF4E (anti-eIF4AI-antibody, 1:100 dilution, product# 2490; anti-eIF4E-antibody, 1:100 dilution, product# 9742; both Cell Signaling Technology, Danvers, MA, USA) was performed on 3 µm-thick sections of NB and NNT specimens using the VENTANA BenchMark XT system and the ultraView Universal DAB detection kit (Ventana Medical Systems, Inc., Oro Valley, AZ, USA). For epitope retrieval, CC1 mild was used. Cytoplasmic staining was evaluated by J.H. and P.C. and the staining intensity was scored as follows: score 0 was assigned to no staining, score 1 to weak, score 2 to moderate, and score 3 to strong staining.

### 2.5. Protein and RNA Isolation

For protein isolation from cryo-tissue, the MagNA Lyser (Roche Diagnostics, Rotkreuz, Switzerland) was used. Nonidet P-40 based buffer (50 mM Tris-HCl, 150 mM NaCl, 0.5% NP-40) was supplemented with 1 mM Pefabloc^®^ SC, 1 mM DTT, cOmplete^TM^ (EDTA-free protease inhibitor cocktail, one tablet for 10 mL lysis buffer), and PhosSTOP^TM^ (phosphatase inhibitor cocktail, one tablet for 10 mL lysis buffer). The finalized lysis buffer was added to the tissue and the mixture was homogenized (6500 rpm for 40 s twice, with 30 s cooling time on ice in between). For the protein isolation of cell lines, cells were scraped off and lysed in finalized lysis buffer via harsh pipetting and vortexing. Protein concentrations were measured using the Bradford protein assay (Bradford solution, Bio-Rad Laboratories, Hercules, CA, USA). RNA was extracted using guanidinium thiocyanate-phenol-chloroform extraction (TRIzol^TM^ Reagent, Invitrogen, Carlsbad, CA, USA) from half of the tissue homogenate or from cell line pellets. After phase separation, RNA precipitation, and RNA washing, the RNA quality and quantity were assessed using the NanoDrop™ 1000 Spectrophotometer (Thermo Fisher Scientific, Waltham, MA, USA).

### 2.6. Western Blot

Semi-dry immunoblotting was performed as described previously [24]. Briefly, 30 µg of total protein lysate was separated using SDS polyacrylamide gel electrophoresis and proteins were blotted on Immobilon-P PVDF membranes (0.45 µm; Millipore, Burlington, MA, USA). For the assessment of whether the protein transfer was successful, membranes were reversibly stained using Ponceau S (Sigma Aldrich, St. Louis, MO, USA). Primary antibodies (anti-eIF4E-antibody product# 9742; anti-eIF4AI-antibody product# 2490; anti-GAPDH-antibody product# 2118; all three Cell Signaling Technology; anti-Actin-antibody product# A2103, Sigma Aldrich) were diluted 1:1000 in Tris-buffered saline supplemented with 0.1% Tween (TBS-T) containing 5% bovine serum albumin (BSA fraction V, Roche Diagnostics) and incubated overnight at 4 °C. Subsequently, horseradish peroxidase conjugated secondary anti-rabbit antibody (1:5000 dilution in 5% non-fat dried milk in TBS-T; ECL™ Anti-rabbit IgG HRP; GE Healthcare, Chicago, IL, USA) was incubated for 1 h at room temperature. For puromycin detection, anti-puromycin antibody (clone 12D10, product# MABE343; MERCK KGaA, Darmstadt, Germany) was diluted 1:5000 in TBS-T containing 5% bovine serum albumin and incubated overnight at 4 °C. Subsequently, horseradish peroxidase-conjugated secondary anti-mouse antibody (1:3000 dilution in 5% non-fat dried milk in TBS-T; ECL™ Anti-mouse IgG HRP; GE Healthcare) was incubated for 1 h at room temperature.

The visualization of proteins of interest via chemiluminescence using Amersham ECL Select Western blotting detection Reagent (GE Healthcare) was performed using the ImageQuant^TM^ LAS 500 (GE Healthcare). The ImageJ software [25] was used for densitometrical analysis and the signals were normalized to GAPDH and Actin as a loading control.

### 2.7. Quantitative Real-Time PCR (RT-qPCR)

A total of 1 µg of RNA was reversely transcribed using the High-Capacity cDNA Reverse Transcription Kit (Applied Biosystems, Foster City, CA, USA) according to the manufacturer’s instructions. For gene expression analyses, 2.5 ng of cDNA were mixed with the Luna^®^ Universal qPCR Master Mix (New England Biolabs, Ipswich, MA, USA). For *EIF4A1, EIF4E*, and *SDHA*, 200 nM of forward (fwd) and reverse (rev) primers was used (*EIF4A1* fwd: 5′-CGAAATGTTAAGCCGTGGATTCA-3’, *EIF4A1* rev: 5´-CTCAAGCACATCAGAAGGCAT-3´; *EIF4E* fwd: 5´-TGCGGCTGATCTCCAAGTTTG-3´; *EIF4E* rev: 5´-CCCACATAGGCTCAATACCATC-3´; *SDHA* fwd: 5´-TGGTTGTCTTTGGTCGGG-3´; *SDHA* rev: 5´-GCGTTTGGTTTAATTGGAGGG-3´; all Eurofins Genomics, Ebersberg, Germany). Besides *SDHA*, *IPO8* was used as a housekeeping gene (QuantiTect Primer Assays, Hs_IPO8_1_SG; final primer mixture concentration 1x, QIAGEN, Hilden, Germany). Gene expression was evaluated with the QuantStudio™ 7 Flex Real-Time PCR System (Applied Biosystems), whereas the run profile was the following: 2 min at 50 °C, 10 min at 95 °C, 45 cycles with 15 s at 95 °C, and 1 min at 60 °C. The relative gene expression was calculated using the 2^−ΔΔCt^ method [26], whereas the expression was measured in triplicate for each sample.

### 2.8. Survival Analysis

Publicly accessible RNA-seq expression data were used to investigate the genes of interest in another neuroblastoma dataset. For the analysis, the RNASeq data of 141 NB patients generated within the Therapeutically Applicable Research to Generate Effective Treatments (https://ocg.cancer.gov/programs/target) initiative (Study ID phs000218) were used (Neuroblastoma sub-study ID phs000467, led by John M. Maris (Children’s Hospital of Philadelphia; Principal Investigator), Robert Seeger (Children’s Hospital of Los Angeles; Co-Principal Investigator), and Javed Khan (National Cancer Institute, Center for Cancer Research; NCI Intramural Lead Investigator)). The data used for this analysis are available at https://portal.gdc.cancer.gov/projects. All the analyses were performed using R version 4.0.2 [27]. For the data retrieval, the Bioconductor package TCGAbiolinks was used [28]. Genes with a minimum median count of 10 were included and normalized as counts per million using the Bioconductor package edgeR [29]. The data were stratified by median expression to identify the association between *EIF4A1* expression, *MYCN*-amplification status (amplified versus non-amplified), and tumor localization (adrenal versus non-adrenal) and OS. The OS was assessed using the log-rank test of the survival package [30]. The package was used to generate Kaplan–Meier survival curves and to fit a cox proportional hazard regression model.

### 2.9. Cell Culture

SH-SY5Y cells were cultured in DMEM/F-12 HEPES (1:1) supplemented with 10% fetal bovine serum (FBS), 100 U/mL penicillin, and 100 µg/mL streptomycin (Gibco^®^, Carlsbad, CA, USA). Kelly cells were cultured in RPMI-1640 supplemented with 10% FBS, 100 U/mL penicillin, 100 µg/mL streptomycin, and 2 mM L-Glutamine (Gibco^®^). Culture conditions were 37 °C in humid atmosphere with 5% CO_2_. Cells were passaged when 90% confluency was reached. Routine check for mycoplasma contamination was conducted using the PCR Mycoplasma Test Kit I/C (PromoCell, Heidelberg, Germany) and STR profiling (PowerPlex 16HS System, Promega, Madison, WI, USA) was performed to verify cell lines.

### 2.10. Cell Viability

For viability assessment and IC_50_ evaluation, SH-SY5Y cells (1.5 × 10^4^ cells/well) and Kelly cells (3 × 10^4^ cells/well) were seeded into 96-well culture plates (clear plate and bottom), allowed to adhere for 24 h, and treated with CR-1-31-B (concentration range 0.1–100 nM) for 24, 48, and 72 h. After incubation, the viability of cells was determined using 3-(4,5-dimethylthiazol-2-yl)-2,5-diphenyltetrazolium bromide (MTT) (Sigma-Aldrich). MTT in DPBS (Dulbecco’s Phosphate buffered saline, Gibco^®^) was added to each well (final concentration 0.55 mg/mL) and incubated for 2 h at 37 °C in a humid atmosphere with 5% CO_2_. Supernatant was discarded, the cells were lysed with 3% SDS, and formazan crystals were dissolved in 40 mM of isopropanol/HCl under vigorous shaking for 15 min. Absorption was measured at 570 nm with the Synergy^TM^ 4 (BioTek^®^, Winooski, VT, USA). Three independent experiments were performed, whereby sextuplicates were used for each condition.

### 2.11. Cell Proliferation

SH-SY5Y cells (1.5 × 10^4^ cells/well) and Kelly cells (3 × 10^4^ cells/well) were seeded into 96-well culture plates (clear plate and bottom) and allowed to adhere for 24 h. SH-SY5Y were treated with 10, 20, and 50 nM of CR-1-31-B, Kelly with 1, 4, and 10 nM of CR-1-31-B for 24, 48, and 72 h. For proliferation assessment, bromodeoxyuridine (BrdU, BrdU Cell Proliferation ELISA kit, abcam^®^, Cambridge, UK) was added in the final 24 h of incubation. ELISA was performed according to the manufacturer’s instructions, with the exception that the peroxidase reaction was stopped after 20 min. The absorption at 450 nm and 550 nm was measured with the Synergy^TM^ 4 (BioTek^®^). Three independent experiments were performed, whereby triplicates were used for each condition.

### 2.12. Apoptosis Assays

The apoptotic behavior of cells when treated with CR-1-31-B was measured using Annexin/Propidium iodide (PI) staining (APC Annexin V Apoptosis Detection Kit with PI, BioLegend^®^, San Diego, CA, USA) and the detection of caspase-3 and -7 activity (Caspase-Glo^®^ 3/7 assay, Promega).

For the Annexin/PI staining, SH-SY5Ycells (2 × 10^5^ cells/well) and Kelly cells (3 × 10^5^ cells/well) were seeded in 6-well plates and allowed to adhere for 24 h. SH-SY5Y cells were treated with 10, 20, and 50 nM of CR-1-31-B and Kelly with 1, 4, and 10 nM of CR-1-31-B for 24, 48, and 72 h. After inhibitor incubation, the supernatant was collected and the cells were detached and pooled with the supernatant. Cells were centrifuged and washed twice with Bio Legend’s cell staining buffer and finally resuspended in 100 µL of Annexin V Binding Buffer, followed by the addition of 2.5 µL APC Annexin V and 5 µL PI and 15 min incubation at room temperature under dark conditions. The addition of 400 µL Annexin V Binding Buffer followed and the samples were analyzed on a CytoFLEX S flow cytometer (Beckman Coulter, Brea, CA, USA). Three independent experiments were performed.

For the Caspase-Glo^®^ 3/7 assay, SH-SY5Y cells (1.5 × 10^4^ cells/well) and Kelly cells (3 × 10^4^ cells/well) were seeded into 96-well culture plates (white plate) and allowed to adhere for 24 h. SH-SY5Y cells were treated with 10, 20, and 50 nM of CR-1-31-B and Kelly with 1, 4, and 10 nM for 24, 48, and 72 h. Afterwards, the supernatant was removed, followed by the addition of 50 µL of fresh media and 50 µL of Caspase-Glo^®^ 3/7 Reagent per well. After 1 h incubation at 37 °C in a humid atmosphere with 5% CO_2_, the luminescence was measured with the Synergy^TM^ 4 (BioTek^®^). Three independent experiments were performed, whereby quadruplicates were used for each condition.

### 2.13. Cell Cycle

SH-SY5Y cells (4 × 10^5^ cells/well) and Kelly cells (6 × 10^5^ cells/well) were seeded in 6-well plates and allowed to adhere for 24 h. SH-SY5Y were treated with 10, 20, and 50 nM of CR-1-31-B and Kelly with 1, 4, and 10 nM of CR-1-31-B for 24, 48, and 72 h. Cells were detached, washed twice with DPBS, fixed with ice cold 70% ethanol, and stored until further usage (no longer than 1 week). Prior staining samples were centrifuged, again washed twice, and then stained with hypotonic PI lysis buffer (0.1% sodium citrate, 0.1% Triton X-100, 100 μg/mL RNAse A, and 50 μg/mL PI) for 20 min at room temperature under dark conditions. Samples were measured with a CytoFLEX LX flow cytometer (Beckman Coulter) and analyzed using ModFit LT^TM^ Version 5.0 (Verity Software House, Topsham, ME, USA). Four independent experiments were conducted.

### 2.14. Puromycin Labelling

For the assessment of efficient translation inhibition by CR-1-31-B via puromycin labelling, SH-SY5Y cells (2.5 × 10^5^ cells/well) and Kelly cells (5 × 10^5^ cells/well) were seeded into 6-well culture plates and allowed to adhere for 24 h. Afterwards, the cells were treated for 24, 48, and 72 h with DMSO (solvent control, 0.5% DMSO) or CR-1-31-B (SH-SY5Y 20 nM; Kelly 4 nM). As a positive control for translation inhibition, untreated cells were treated with cycloheximide (50 µg/mL) for 5 h. Ten minutes before cell harvest, cells were treated with 10 µg/mL of puromycin dihydrochloride (Gibco^®^). All the incubation steps were at 37 °C in a humid atmosphere with 5% CO_2_. After incubation, the supernatant was discarded and the cells were washed twice with ice-cold DPBS and scraped off. After protein isolation, semi-dry immunoblotting was performed.

## 3. Results

### 3.1. eIF4AI Is Significantly Overexpressed at the Protein and mRNA Level in FFPE and Fresh Frozen Human Neuroblastic Tissue Compared to Non-Neoplastic Tissue

The overexpression of eIF4AI and eIF4E has been documented in different cancer types. Therefore, the expression levels of these factors were evaluated at the protein and mRNA level in neuroblastic (NB) tissue.

Immunohistochemical protein expression analysis was performed on the tumor tissue of 36 NB patients and compared to NNT. Representative immunohistochemical stainings with no, weak, moderate, and strong eIF4AI and eIF4E staining intensity of NB tissue and no staining of NNT are shown in Figure 1A and Appendix A. The significant overexpression of eIF4AI (*p* < 0.001) and eIF4E (*p* < 0.001) was seen in tumor tissue compared to NNT. In total, 72% (47% weak, 17% moderate, and 8% strong staining) of NB patients showed positive eIF4AI immunoreactivity (Figure 1B). eIF4E immunoreactivity was found in 47% of the NB patients (39% weak, 8% moderate staining) (Appendix A). Although NB is biologically very heterogenous, a difference in eIF4AI and eIF4E expression was absent in this cohort when stratified for adrenal and non-adrenal localization (*p* = 0.285 and *p* = 1.000, respectively), as well as for the *MYCN* amplification status (*p* = 0.689 and *p* = 0.717, respectively) of the tumor. Additionally, a significant difference in eIF4AI and eIF4E expression was absent in this cohort when stratified based on tumor stages 1–2 versus 3–5 (eIF4AI: *p* = 1.000; eIF4E: *p* = 0.451).

For the protein expression analysis, the cryo-tissue of NB patients (n = 17) and NNT was analyzed using immunoblotting. Figure 1C shows representative immunoblots for eIF4E, eIF4AI, actin, and GAPDH. Densitometric evaluation of all samples (Figure 1D) showed the significant (*p* < 0.001) overexpression for eIF4AI and eIF4E normalized to the loading control in NB tissue compared to NNT. Gene expression analysis revealed the significant overexpression of *EIF4A1* (*p* < 0.001), whereas the *EIF4E* mRNA expression was unaffected (*p* > 0.5) in NB compared to NNT tissue.

The influence of *EIF4A1* overexpression on patient OS was evaluated using the publicly available dataset of the Therapeutically Applicable Research to Generate Effective Treatments (TARGET) initiative (clinicopathological data of 141 NB patients with an available MYCN status were included; see Appendix A). Appendix A shows the Kaplan–Meier curves based on high versus low *EIF4A1* expression, *MYCN* amplification status (amplified versus non-amplified), and localization (adrenal versus non-adrenal). When stratified by median *EIF4A1* expression, the *EIF4A1* expression status was not significantly associated with OS. Neither *MYCN* amplification status nor tumor localization significantly impacted OS, with the exception of having the tumor localized in the adrenal gland and high *EIF4A1* expression without *MYCN* amplification versus adrenal tumor and low *EIF4A1* expression plus *MYCN* amplification. The OS was significantly lower (*p* = 0.0086) when having low *EIF4A1* expression plus *MYCN* amplification (HR = 3.55 [95% CI: 0.79-16.01] *p* = 0.1).

### 3.2. Targeting eIF4AI with CR-1-31-B Reduces Viability, Proliferation and Changes Cell Cycle Phase Distribution at Low nM Doses

Significant eIF4AI overexpression was observed for the protein and mRNA levels in NB tissue compared to NNT. To evaluate the potential of targeting eIF4AI in NB, the NB cell lines SH-SY5Y and Kelly were treated with the rocaglate derivative, CR-1-31-B.

Cell lines were treated with various CR-1-31-B concentrations. Cell viability was assessed 24, 48, and 72 h after compound exposure. A significant decrease in SH-SY5Y viability upon CR-1-31-B was observed at 10 nM for all time points (24 h *p* < 0.05; 48 h and 72 h *p* < 0.001) in comparison to the vehicle control (solvent control, 0.5% DMSO, VC) (Figure 2A). CR-1-31-B (5 nM) was sufficient to significantly decrease the viability of Kelly cells (*p* < 0.001) (Figure 2B). The calculated IC_50_ for CR-1-31-B at 48 h was 20 nM for SH-SY5Y and 4 nM for Kelly cells. A significant decrease in the proliferation of SH-SY5Y compared to VC was observed starting from 10 nM for all time points (24 h *p* < 0.01; 48 h and 72 h *p* < 0.001) (Figure 2C). For Kelly cells, the proliferation index was significantly (*p* < 0.05) decreased after 24 h at a concentration of 10 nM of inhibitor. For 48 h and 72 h, a concentration of 5 nM was sufficient for significant proliferation index reduction (*p* < 0.001) (Figure 2D).

To assess the efficacy of translation inhibition by CR-1-31-B, puromycin labelling was used. Cells were treated with calculated IC_50_ of CR-1-31-B for 24, 48, and 72 h; VC; or cycloheximide, which served as a positive control for translation inhibition. Appendix A show the immunoblots for puromycin of SH-SY5Y and Kelly cells, respectively, as well as the appropriate Ponceau S staining, which served as a loading control. No puromycin labelling was observed when SH-SY5Y cells were treated with cycloheximide. In Kelly cells, cycloheximide treatment led to a puromycin labelling reduction of 80%. Compared to VC, puromycin labelling in SH-SY5Y was reduced by 90% and in Kelly by 50% when treated with CR-1-31-B, independent of incubation time.

To investigate the cell cycle phase changes, cells were treated with different concentrations of CR-1-31-B for 24 h and 48 h, followed by PI staining. The proportion of SH-SY5Y in the G_0_/G_1_ phase was significantly reduced when the cells were treated for 24 h with 20 nM (*p* < 0.05) or 24 h and 48 h with 50 nM (*p* < 0.001) CR-1-31-B. The number of SH-SY5Y cells in G_2_/M increased when using 50 nM of inhibitor (24 h, *p* < 0.05; 48 h *p* < 0.01) (Figure 2E). For Kelly cells, the proportion of cells in the G_0_/G_1_ phase increased when treated with 4 nM or 10 nM of CR-1-31-B for 24 h (*p* < 0.001) and 48 h (*p* < 0.01), along with a significant decrease in cells in the S phase (*p* < 0.001) (Figure 2F).

### 3.3. CR-1-31-B Treatment Leads to Apoptotic Cell Death of SH-SY5Y and Kelly Cells

To evaluate whether eIF4AI blockage induces cell death via apoptosis, cells were treated with various CR-1-31-B concentrations, followed by the assessment of apoptosis using two different assays (Annexin V/PI uptake and caspase-3/-7 activity).

The percentage of viable SH-SY5Y cells was unaffected by CR-1-31-B treatment (at 24 h). A significant decrease in viable cells and an increased number of late apoptotic cells (*p* < 0.001) was observed in the presence of 20 nM and 50 nM inhibitor after 48 h and 72 h in SH-SY5Y cells (Figure 3A). The viability of Kelly cells was also reduced following 24 h exposure to 10 nM of compound (*p* < 0.05). A significant (*p* < 0.001) decrease in viable cells and a significantly (*p* < 0.001) increased number of late apoptotic cells were detected using 4 nM or higher concentrations of CR-1-31-B after 48 h and 72 h (Figure 3B).

Increased caspase-3/-7 activity was measured using 20 nM or higher concentrations of CR-1-31-B for 48 h and 72 h (48 h 20 nM and 50 nM *p* < 0.001; 72 h 20 nM *p* < 0.01; 72 h 50 nM *p* < 0.001) in SH-SY5Y cells (Figure 3C). For Kelly cells, increased activity was detected after 48 h at 4 nM (*p* < 0.05) and 48 h 10 nM (*p* < 0.001) inhibitor (Figure 3D).

## 4. Discussion

NB represents a neoplasm with high biological heterogeneity. Aggressive metastatic tumors often remain unresponsive to pharmacological anti-cancer strategies. Despite aggressive multimodal therapy, long-term OS rates are below 50% and new therapeutic targets are urgently needed [2]. Protein synthesis is often dysregulated in cancer. In the past few decades, extensive research, particularly in translation initiation, has revealed that the eIF4F complex may be the Achilles heel common to various tumor entities [31].

To the best of our knowledge, this study presents the first evidence of eIF4AI overexpression in NB. Especially cancer cells highly depend on eIF4AI dependent mechanisms of translational control. Many oncogenes (such as *MYC* or *BCL2*) have been shown to require the helicase for translation due to their complex secondary structure of the 5′ UTR (untranslated region) [32]. This study shows that targeting the translation machinery *in vitro* with the synthetic rocaglate CR-1-31-B may be an effective way to target NB cells at concentrations that non-malignant cells have not been affected by in previous studies.

The role of dysregulated eIFs in NB is largely unknown. Parker et al. demonstrated that eIF4E expression was significantly higher in NB patients below the age of 12 months, but failed to show any significant correlation of staining intensity and tumor stage or mortality [23]. In accordance with their findings, we are able to show the overexpression of eIF4E at the protein level in NB tissue, but there was no difference at the mRNA expression level. When having unchanged mRNA expression, elevated protein levels may be explained by increased translation rates or increased protein stability in tumor tissue. eIF4E was shown to be rate-limiting for translation, since it is the scarcest eIF [8]. Increased demand for cap-dependent translation and thus levels of eIF4E might drive cancer cells to change the eIF4E protein stability or eIF4E mRNA translational efficiency. Previous studies have indicated that eIF4AI is overexpressed in different tumor entities. Liang et al. reported that eIF4AI overexpression is an acquired phenotypic feature in cervical cancer and that it might function as a prognostic indicator as well as therapeutic target [14]. In the study by Comtesse et al., *EIF4A1* overexpression was demonstrated in lung cancer [13]. For the first time, we demonstrate that eIF4AI is also overexpressed at protein as well as at mRNA level in human NB tissue compared to NNT. This finding is independent of whether the tissue originated from patients with ganglioneuroblastoma or NB, tumor localization, or MYCN amplification status, Additionally, *EIF4A1* overexpression did not affect the patient’s OS (dataset from TARGET initiative).

Rocaglates primarily target eIF4AI in yeast [33] and mammalian cells [20] and are responsible for the inhibition of translation. One prominent rocaglate is silvestrol, which was shown to be tumor-suppressive in breast and prostate cancer xenografts [19]. The synthesis of silvestrol is challenging. By preparing cyclopenta[*b*]benzofuran analogous without the dioxanyl functional group, one compound, CR-1-31-B, was found to be as potent as silvestrol, as shown in human Burkitts’ lymphoma BJAB cells and in a lymphoma mouse model [22]. A recent study by Chan and colleagues found that CR-1-31-B treatment reduced the viability of pancreatic ductal adenocarcinoma (PDAC) organoids and suppressed tumor growth in a rodent PDAC model [34]. We confirmed that CR-1-31-B targets translation using puromycin labelling and show that the viability of the NB cell lines is reduced by 50% already at low nM doses (4 nM for Kelly, 20 nM for SH-SY5Y). This is in accordance with the use of 10 nM in the PDAC organoid study [34] and 0.5–20 nM in BJAB cells [22]. Rust et al. have demonstrated that the eIF4AI inhibiting bacterial toxin burkholderia lethal factor 1 (BLF1) is very potent against NB cell lines with an amplification of *MYCN* [35]. This is consistent with our study, since CR-1-31-B is cytotoxic at lower concentration for Kelly cells (100-fold genomic amplification of *MYCN* [36]) compared to SH-SY5Y (no *MYCN* amplification). Besides reducing viability, CR-1-31-B is also capable of curtailing the proliferation of NB cells and induces apoptotic cell death. The cell cycle behavior under CR-1-31-B treatment differs between the two tested NB cell lines. While CR-1-31-B increases the number of SH-SY5Y cells in the G_2_/M-phase and decreases the amount of cells in the G_0_/G_1_-phase, the proportion of Kelly cells in G_0_/G_1_ increases while the number of cells passing into S-phase decreases. Our results indicate that, during treatment with CR-1-31-B, Kelly cells undergo G_0_/G_1_ arrest. This is in accordance with the results in breast and non-small cell lung cancer cells by Kong et al., showing that resistance to CDK4/CDK6 inhibition via the induced feedback upregulation of Cyclin D1 and CDK4 is reversible using 3.2 nM of CR-1-31-B [37].

The results of the CR-1-31-B treatment of the two tested NB cell lines revealed that targeting eIF4AI may present a therapeutic strategy for NB independent of *MYCN* status. whose amplification correlates with advanced disease stages [38].

Ever since screening studies for biological activities in the 1970s revealed that substances produced by the *Aglaia* sp have anti-neoplastic properties [39], rocaglates have been of major interest in cancer research. The best characterized rocaglate family member named silvestrol has been shown to have promising cytotoxicity against different cancer cell lines [21]. As mentioned above, the chemical synthesis of silvestrol is challenging and, in this regard, the synthetic hydroxamate derivative CR-1-31-B was found to exhibit equal anti-neoplastic activity, with the advantage of reduced complexity in chemical synthesis. It was also shown to be biologically stable in human plasma and to exhibit significant hepatic stability [22]. We are sensible about the fact that this study lacks results on matching non-malignant cell lines, emphasizing the specificity of CR-1-31-B against NB. In this respect, Robert and coworkers were able to show that silvestrol is effective against a variety of multiple myeloma cell lines at low nM doses but is cytotoxic for non-malignant cell lines only at concentrations greater 100 nM [40]. This finding indicates the specificity of silvestrol against cancerous cells compared to non-malignant cells. For CR-1-31-B, a derivative of silvestrol, Chan and colleagues could show that it effects pancreatic organoids at doses comparable to our study, with effects on non-cancerous organoids only seen at tenfold higher IC_50_ concentrations [34], highlighting the higher activity of CR-1-31-B against cancerous cells.

We are aware that our study faces some limitations. One weakness was the NNT obtained from brains and spinal cords and derived from autopsies from adults. NB is a pediatric tumor and tissue availability is an issue. Furthermore, an increase in NB cases would reinforce our findings, as well as treatment studies in mice- and patient-derived tumor cells with matching fibroblasts. A recent study by Chu et al. identified an even more potent class of rocaglates—the amidino-rocaglates (ADRs) [41]. The leading compound in this study, CMLD012612, was found to be more efficient in inhibiting translation and reducing viability in NIH/3T3 cells compared to CR-1-31-B (IC_50_ of 2 nM versus 9 nM). Additionally, the new compound showed synergizing effects with doxorubicin in mice with lymphoma, such as CR-1-31-B [22], and found that ADRs are effective in targeting eIF4A in vitro and in vivo [41]. Future studies will be necessary to evaluate the potency of ADRs in NB.

## 5. Conclusions

In summary, our results clearly demonstrate for the first time that eIF4AI is overexpressed in NB tissue compared to NNT. Targeting the translational machinery in NB in vitro is effective, thus opening new opportunities for NB therapy.

## Figures and Tables

**Figure 1 cells-10-00301-f001:**
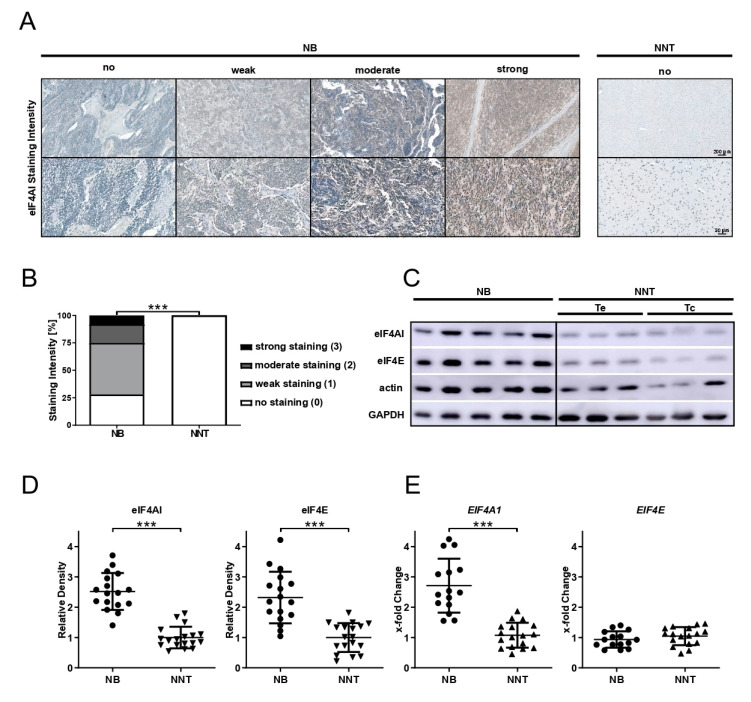
Protein and mRNA expression analysis of eIF4AI and eIF4E in neuroblastic (NB) and non-neoplastic tissue (NNT) reveals the overexpression of eIF4AI in NB compared to NNT. (**A**) Representative immunohistochemical staining with no, weak, moderate, and strong eIF4AI immunoreactivity of NB tissue and no staining of NNT. Scale bars: 200 µm for upper row and 50 µm for bottom row. (**B**) Comparison of the immunohistochemical eIF4AI staining intensity of NB (n = 36) and NNT (n = 18) using the Mann–Whitney U test. *** *p* < 0.001; no staining (white bar) NB n = 10, NNT n = 18; weak staining (light grey bar) NB n = 17; moderate staining (dark grey bar) NB n = 6; strong staining (black bar) NB n = 3. (**C**) Representative immunoblots for eIF4AI, eIF4E, actin, and GAPDH expression in NB and NNT. Te, temporal cortical brain tissue; Tc, thoracic spinal cord tissue. (**D**) Relative density values of eIF4AI and eIF4E expression of each sample [NB (n = 17) and NNT (n = 18)] determined by immunoblotting normalized to average of loading control (actin and GAPDH). Densitometrical analysis was performed using ImageJ. Unpaired *t*-test was used for statistical analysis, bars represent mean ± SEM, *** *p* < 0.001. (**E**) Gene expression analysis (qRT-PCR) of *EIF4A1* and *EIF4E* in NB (n = 17) compared to NNT (n = 18). Data are shown as x-fold change using the 2^−ΔΔCt^ method. Mean of *IPO8* and *SDHA* served as endogenous controls. Unpaired t-test was used for statistical analysis, bars represent mean ± SEM, *** *p* < 0.001.

**Figure 2 cells-10-00301-f002:**
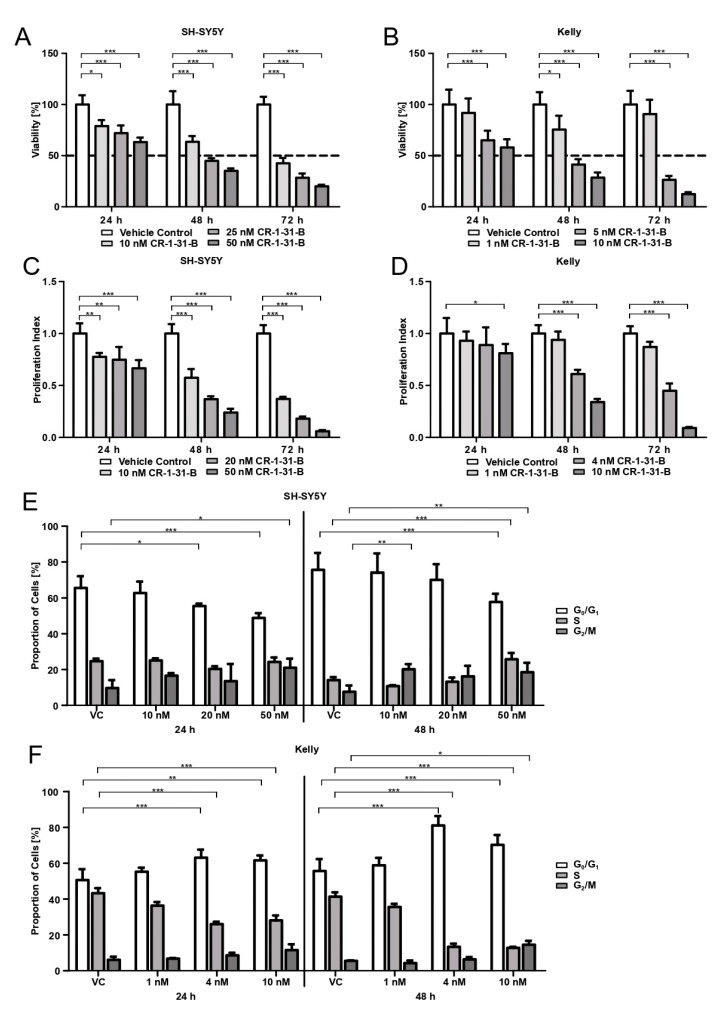
Targeting eIF4AI using rocaglate derivative CR-1-31-B in neuroblastoma cell lines SH-SY5Y and Kelly reduces cell viability and proliferation with cell cycle phase alterations over time. SH-SY5Y cells were treated with 10, 20 (25 nM for viability), and 50 nM and Kelly cells with 1, 5, and 10 nM CR-1-31-B over 24, 48, and 72 h. As a vehicle control, 0.5% (v/v) DMSO was used (VC). A and B show data of the viability changes upon CR-1-31-B treatment using MTT assay of SH-SY5Y cells (**A**) and Kelly cells (**B**), represented as the percentage of viable cells normalized to VC. C and D show alterations in the proliferation index upon CR-1-31-B treatment using BrdU assay in SH-SY5Y cells (**C**) and Kelly cells (**D**) normalized to VC. Determination of cell cycle phases of SH-SY5Y cells (**E**) and Kelly cells (**F**) upon CR-1-31-B treatment was measured by flow cytometry. Depicted are percentages of cells in the G0/G1 phase (white bars), S-phase (light grey bars), and G2/M-phase (dark grey bars). Two-way analysis of variance with Bonferroni post-hoc-test was used for statistical analysis for all tests, bars represent mean ± SEM, * *p* < 0.05, ** *p* < 0.01, *** *p* < 0.001, three independent experiments were performed for MTT and BrdU assay, cell cycle was evaluated in four independent experiments.

**Figure 3 cells-10-00301-f003:**
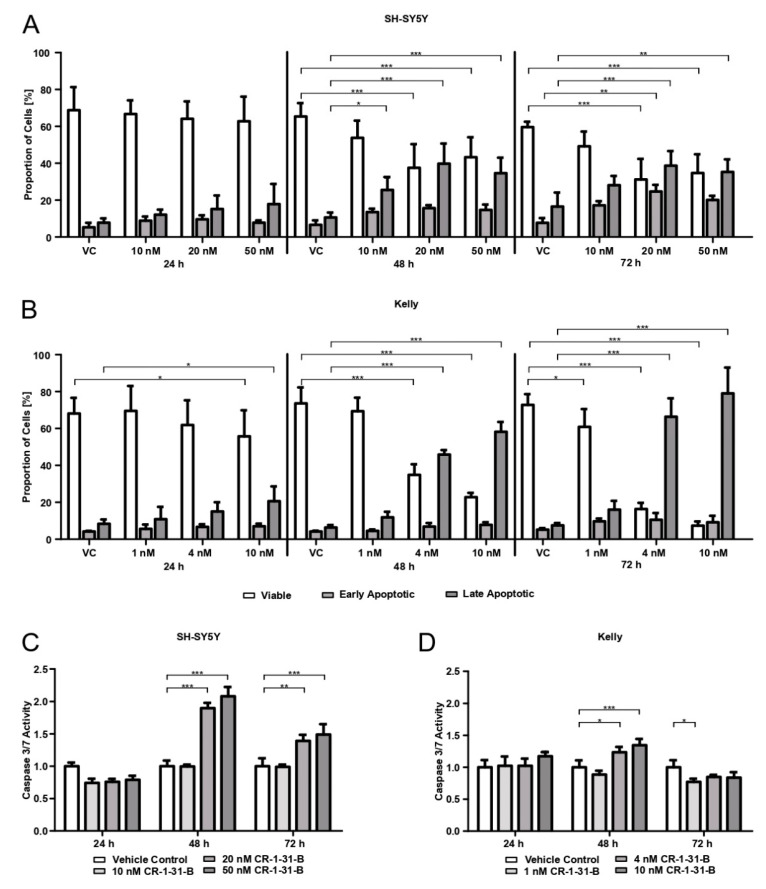
Targeting eIF4AI using rocaglate derivative CR-1-31-B in neuroblastoma cell lines SH-SY5Y and Kelly triggers apoptosis. SH-SY5Y cells were treated with 10, 20, and 50 nM and Kelly cells with 1, 5, and 10 nM CR-1-31-B over 24, 48, and 72 h. As a vehicle control, 0.5% (v/v) DMSO was used (VC). Apoptosis was evaluated using Annexin (Ax)/Propidium iodide (PI) staining (**A**: SH-SY5Y cells; **B**: Kelly cells) and the detection of caspase-3/-7 activity (**C**: SH-SY5Y cells, **D**: Kelly cells). (**A**,**B**) show the proportion of cells that are viable (white, Ax-/PI-), early apoptotic (light grey, Ax+, PI-), and late apoptotic (dark grey, Ax+/PI+). (**C**,**D**) show the activity of Caspase-3 and -7 in relation to VC. Two-way analysis of variance with Bonferroni post-hoc-test was used for statistical analysis for both tests, bars represent mean ± SEM, * *p* < 0.05, ** *p* < 0.01, *** *p* < 0.001, three independent experiments were performed for both tests.

**Table 1 cells-10-00301-t001:** Clinicopathological data of patient-derived neuroblastoma tissue and warm autopsy-derived non-neoplastic tissue.

	Tissue for Biochemical Analyses (n = 17)	NB Tissue for IHC Analyses (n = 36)	NNT for IHC and Biochemical Analyses (n = 18)
**Age (Median)**	15 m	27 m	61 y
**Gender**			
**Female**	7 (41%)	15 (42%)	4 (22%)
**Male**	10 (59%)	21 (58%)	14 (78%)
**Stage**			
**1**	1 (6%)	5 (14%)	
**2**	3 (18%)	4 (11%)	
**3**	4 (24%)	13 (36%)	
**4**	3 (18%)	12 (33%)	
**5**	0 (0%)	2 (6%)	
**Unknown**	6 (35%)	0 (0%)	
**MYCN Status**			
**Non-amplified**	13 (76%)	26 (72%)	
**Amplified**	2 (12%)	10 (28%)	
**Gain**	1 (6%)	0 (0%)	
**Unknown**	1 (6%)	0 (0%)	
**Localization**			
**Adrenal gland**	4 (24%)	20 (56%)	
**Retroperitoneal**	1 (6%)	8 (22%)	
**Mediastinal**	0 (0%)	8 (22%)	
**Abdominal**	3 (18%)	0 (0%)	
**Thoracic**	4 (24%)	0 (0%)	9 (50%)
**Others**	5 (29%)	0 (0%)	
**Temporal lobe**	0 (0%)	0 (0%)	9 (50%)
**Diagnosis**			
**Ganglioneuroma**	0 (0%)	4 (11%)	
**Ganglioneuroblastoma**	6 (35%)	5 (14%)	
**Neuroblastoma**	11 (65%)	27 (75%)	
**Differentiated**	n.d.	13 (48%)	
**Poorly differentiated**	n.d.	12 (44%)	
**Undifferentiated**	n.d.	2 (7%)	

m = months; y = years; n.d., not defined; NB = neuroblastoma; NNT = non-neoplastic tissue, IHC = immunohistochemical.

## Data Availability

RNASeq used for survival analyses was generated within the Therapeutically Applicable Research to Generate Effective Treatments initiative (Study ID phs000218) (Neuroblastoma sub-study ID phs000467, led by John M. Maris (Children’s Hospital of Philadelphia; Principal Investigator), Robert Seeger (Children’s Hospital of Los Angeles; Co-Principal Investigator), and Javed Khan (National Cancer Institute, Center for Cancer Research; NCI Intramural Lead Investigator)) and is publicly available at https://ocg.cancer.gov/programs/target.

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
