# Peer review of "Eukaryotic Translation Initiation Factor 4AI: A Potential Novel Target in Neuroblastoma"

_cells, 2021, doi:10.3390/cells10020301_

Round 1
Reviewer 1 Report
The manuscript by Christina Skofler et al., has potentially interesting findings. It shows that NB tumors have increased eIF4A1 and eIF4E levels as compared to adult brain and spinal cord samples (obtained from autopsies). The authors then show that CR-1-31-B, an inhibitor of eIF4A has profound growth and protein synthesis inhibitory effects in NB cell lines and it can induce apoptosis in higher concentrations. Based on these findings, the authors propose CR-1-31-B inhibitors for NB therapeutics.
Comments
The presented data are consistent with the conclusions of the authors. However, it is very unclear what is different in the NB cell lines as compared to more normal non tumorigenic cells exposed to the same concentrations of the drug. The authors should consider testing in parallel the same outcomes as in Figs 2 and 3, in other cell lines and identify a unique nature of NBs. This unique nature can be that the effects may be observed in NBs at lower drug concentrations. This may support then the data of Fig. 1, by correlation. Unless a comparison with other relevant cells is made, I cannot see a great value in this study. For example, a pair in breast cancer cells, can be MCF10A cells. The authors should consider a pair for NBs and demonstrate a difference in the response to CR-1-31-B.
Reviewer 2 Report
Skofler and colleagues investigated the levels of eukaryotic initiation factor 4AI (eIF4AI), in neuroblastoma cells. They performed immunohistochemical staining of the formalin-fixed parafilm embedded (FFPE) clinical specimens as well as immunoblotting of the cryo-tissue samples. Despite a high biological heterogeneity of the neuroblastoma cancer cells, strongly increased mRNA and protein levels of the eIF4AI were uniformly observed in contrast to the non-neoplastic tissue.
Authors tested CR-1-31-B (a rocaglate derivative drug) to specifically block the eIF4AI and subsequently evaluate cell viability, cell cycle progression, and susceptibility to apoptosis.
They showed that this drug was effective in nanomolar concentration in vitro in SH-SY5Y and Kelly neuroblastoma cell lines. Both cell lines manifested decreased viability in a dose-dependent manner and the cell death was mostly apoptotic which was confirmed using annexin V and caspase3/7 assays.
Overall, the study is timely and relevant considering the scarcity of publications on the pathology of neuroblastoma and the limitations that scientists could find when studying such a rare pediatric cancer.
The experiments in this manuscript appear to be carefully designed and executed with the appropriate statistical validation. However, the description was quite vague and confusing at times, especially with respect to the interpretation of the experiments related to the cell cycle progression upon treatment with the CR-1-31-B compound.
Lines 347-352: “To investigate cell cycle phase changes, cells were treated with different concentrations of CR-1-31-B for 24 h and 48 h, followed by PI staining. The proportion of SH-SY5Y in the G0/G1 phase was significantly reduced by 20 nM for 24 h (p<0.05) and 50 nM for 24 h and 48 h (p<0.001). The amount of SH-SY5Y in G2/M increased when using 50 nM of inhibitor (24 h, p<0.05; 48 h p<0.01) (Figure 2E). For Kelly cells, the proportion of cells in G0/G1 phase increased by treatment with 4 nM or 10 nM CR-1-31-B at 24 h (p<0.001) and 48 h (p<0.01) along with a significant decrease of cells in the S phase (p<0.001) (Figure 2F)”.
Perhaps the authors meant that when the inhibitor is used at 20 nM, the number of SH-SY5Y cells gets reduced because at this point that sentence simply does not make any sense.
Figure 2: SH-SY5Y in Figure 2C are showing very low proliferation rates at 50nM of CR-1-31-B treatment after 48h, but Figure 2E shows the opposite as the proportion of 50nM treated cells in the G2/M phase is higher than those of VC. And same for Kelly cells. The authors gave the following explanation (Line 426-429):
“SH-SY5Y showed no G0/G1 arrest upon treatment with CR-1-31-B. An explanation for this phenomenon may be the long population doubling time of 50-60 h of SH-SY5Y cells”. This argument is not convincing: an easy way to verify is to run a cell cycle assay for 50nM treatment at 72h. Why the authors specifically in this assay omitted the 72h timepoint?
Study design (Page 3, lines 91-98 and Table 1): the authors are aware of the confounding issues that the control group presents in this study: a) the median age of the NNT control group is 61y while the NB groups have median ages of 15m and 27m. Age is a confounding variable, and to choose a starkly contrasting age category as a control for the study is obviously a significant bias that weakens the whole study. b) the size of the control group is very small. c) gender bias: the ratio M/F in NNT is biased compared to case groups.
Figure 1C: is there any particular reason why the levels of actin are not consistent in NNT samples?
Figure 1E: the authors explain why the higher expression levels of eIF4E protein in NB samples (2.5-fold higher) is not reflected at the RNA levels by the following statement: “increased translation rates or increased protein stability in tumor tissue”. Both explanations are poorly substantiated, and the authors appear to contradict their own statement by saying “Increased demand of cap-dependent translation and thus levels of eIF4E might drive cancer cells to change eIF4E protein or mRNA stability. In this regard, Topisirovic et al. showed that eIF4E mRNA stability was upregulated in cancer cells as a consequence of elevated human antigen R levels” that could simply mean that due to the increased demand, eIF4E mRNA will be either more stable or highly expressed, and in both cases, mRNA levels will be higher.
Also, deciding to “Therefore, in vitro analyses focused on eIF4AI” (Page 9 line 298), is probably a bit hasty, and the authors might need to reconsider the evaluation of eIF4E in the cell lines before drawing any conclusions.
In the discussion, the authors put their results in the context of other findings in cancer and eukaryotic translation research. The cited works are mostly recent, which emphasizes the relevance of the findings in this manuscript. Also, the authors emphasize the awareness of the limitation that the control samples come from adults (neuroblastoma is predominantly a pediatric tumor) and overall sample availability issue.
It is recommended that the authors validate the mRNA and protein levels of eIF4AI in both SH-SY5Y and Kelly lines to demonstrate directly the effect of the CR-1-31-B drug. This should be feasible and also it may help to shed light on why two cell lines alter the cell cycle progression in the opposite manner upon CR-1-31-B treatment. Also, the text needs improvements, because its readability is poor in some parts. There are minor mistakes and typos in the text, which need to be also addressed.
Reviewer 3 Report
The manuscript entitled "Eukaryotic Translation Initiation Factor 4AI: A Potential Novel Target in Neuroblastoma" by Skofler et al. is focused on cytotoxic effects of a small molecule drug, CR-1-31-B, on cultured neuroblastoma cells. CR-1-31-B is a synthetic rocaglate, which is known to clamp translation initiation factor eIF4A (a component of the eIF4F complex) on mRNA during ribosomal loading and scanning, leading to translation repression (with some selectivity towards mRNA species).
In the study, an elevated expression of eIF4AI, at both mRNA and protein levels, was revealed in neuroblastoma cells by IHC, WB and qPCR. Then effects of CR-1-31-B on neuroblastoma cell viability, cell cycle, efficacy of translation and apoptosis rate were estimated, leading to a conclusion that eIF4A targeting can be an anti-tumorogenic approach for neuroblastoma.
Rapidly proliferating cancer cells are known to be highly dependent on active protein synthesis, therefore translation machinery is often regarded as the “Achilles heel” of cancer. A number of small molecules targeting the eukaryotic ribosome or other components of translational apparatus have been engaged in clinical trials. eIF4F/4A inhibitors are of special interest, as they have been shown to somewhat specifically affect a fraction of mRNAs important for rapid proliferation. In this regard, the study by Skofler et al. is promising, interesting and important.
However, it should be kept in mind that protein synthesis is a process that is absolutely essential for any cell, not only cancer ones. Thus, only those drugs have clinical potential that are much more toxic to cancer cells than normal tissues. A good example is a study of cytotoxic effects of another rocaglate, silvestrol, on B-cell acute lymphoblastic leukemia: https://pubmed.ncbi.nlm.nih.gov/19190247/, where a striking difference in IC50 of the drug in the malignant vs. normal cells was shown. Similar approaches were applied also in (https://pubmed.ncbi.nlm.nih.gov/31171817/) and in (https://pubmed.ncbi.nlm.nih.gov/31723131/, btw describing adenocarcinoma treatment with the same drug, CR-1-31-B). The manuscript by Scofler et al. in its current form, unfortunately, provides no evidence that CR-1-31-B is selective for neuroblastoma cells vs. non-malignant ones, since both tested cell lines were tumor cells. It is especially important, as in previous tests (for example, a comprehensive study of the effects of ~200 rocaglates on cell viability, see https://pubmed.ncbi.nlm.nih.gov/32101697/) CR-1-31-B was shown to effectively kill non-tumor cell line, e.g. NIH3T3, in similar ~10-100 nM concentrations.
Thus, to my opinion, this manuscript can be published in Cells only after major revision, if the authors provide additional evidence of a higher CR-1-31-B cytotoxicity toward the neuroblastoma cell lines, as compared to (at least one) non-tumor cell line.
Minor Points:
Although the manuscript is carefully written in general, there are many places where the text could be significantly improved.
- line 33: “High risk advanced… neuroblastoma” should be re-formulated.
- line 37: “eIF4F complex … is the direct link to … signaling pathways” – sounds not good.
- line 80: “inhibition of eIF4G … activity” – the phrase is misleading, as eIF4G has no enzymatic activity (only RNA-binding and protein-binding activity has been documented for this protein); it is likely eIF4F that is meant here.
- Throughout the Intro section, a few references to reviews describing a variety of small molecule inhibitors targeting eukaryotic translation, as well as their clinical application would be helpful (e.g. https://pubmed.ncbi.nlm.nih.gov/25743081/, https://pubmed.ncbi.nlm.nih.gov/32151059/ and/or https://pubmed.ncbi.nlm.nih.gov/33280581/, the latter being probably the broadest one).
- The original paper where CR-1-31-B was developed and tested against lymphoma cells (https://pubmed.ncbi.nlm.nih.gov/22128783/) should be cited either in “Introduction” or in the beginning of section 3.2 of “Results”, as well as probably some other studies where this drug was used against cancer cells (see https://scholar.google.ru/scholar?q=CR-1-31-B, see also above). This is important for understanding the originality of the research by a reader. I would also suggest citing other cases of eIF4A targeting in neuroblastoma (e.g. https://pubmed.ncbi.nlm.nih.gov/29954071/).
- line 120: to my opinion, this way of estimation of the staining intensity (just “evaluated by J.H.”) is too subjective; moreover, it is even more subjective, as there is no internal control (e.g. normal tissue) on the same histological slide – thus, the intensity may depend on accidental differences in the staining procedure in each case. However, I’m not an expert in IHC – so could you provide some evidence that such a way is correct, or replace it with some other method of estimation?
- lines 275-276: please check the sentence.
- line 287: “densities of eIF4AI and eIF4E” – please re-formulate (densities of staining?)
- All figure legends: is it indeed necessary to capitalize letters in the legends?
- Figs. 2 and 3: adding the names of cell lines directly on the figures (in the legends or just above the panels) would improve understanding.
- Fig. 2E: please provide evidence that the differences in the cell cycle phases were not just due to a difference in cell densities (i.e. the cells grown with the drug probably reached a lower density, while those grown with vehicle could reach the 100% confluency and be affected by contact inhibition, therefore more cells could be in G0 there, as we indeed see in the figure): information about cell density in VC would be suitable.
- line 399 – please abbreviate “human antigen R” as HuR, since it is difficult to recognize this protein in such a spelling.
- lines 41, 82/83, 93, 389 and elsewhere – please check paragraph marks.
- lines 96, 104, 292, 368 and elsewhere – please remove extra dots, spaces and other extra symbols.
Round 2
Reviewer 1 Report
I still think that the paper needs to emphasize the reason that the new drug was developed and the value it offers for therapeutics. As the authors explain in their response to reviewers, they should include a paragraph in the Discussion on exactly what is the value that the study brings to the field. Because there is no mechanism and no matching normal cell data, it is essential this to be emphasized. I suggest, second paragraph of the Discussion.
Reviewer 2 Report
I am satisfied with the authors' response to my criticisms and I thank them for their diligence in addressing them.
Reviewer 3 Report
To my opinion, the paper is almost ready to be published. However, the authors’s response to my main concern (which is similar to the main comment of Reviewer 1) should be addressed in the text of the manuscript in a more clear form. It is indeed important to provide arguments (with clear indication in the text) that the drug is more active against cancer cell, as compared to non-tumor ones. Although the data with MRC5 cells cannot be published in the present paper, the authors could mention other arguments came from previous studies. Such arguments have been presented in the author’s response but have not been included in the text of the paper. My opinion is that they should be included, preferentially as a separated paragraph in the Discussion section (e.g. right before limitations, or in some other place). For example, the sentences about “IC50 above 100 nM” (regarding silvestrol) from the authors’ response would be suitable. The fact regarding lymphoma mouse model is also a good argument that can be repeated (or moved) here. Furthermore, a few words about issue should be added into Abstract section (some words indicating that the drug was active against NB in concentrations lower than those shown previously to affect normal cells).
In conclusion, the paper can be accepted without additional review cycle, but the above changes should be made.
To the technical editor:
- Some references need format correction, e.g. spaces between volume numbers are lacking; journal name abbreviations are wrong in 5, 15, 16 and 30.
- Two paragraphs in Abstract look unusual.
